



# Complex analysis of the middle-latitude ionosphere parameters during the geomagnetic storm at Jan, 20, 2010 based on the DEMETER satellite data analysed using DIAS Software

Anatoliy Lozbin[1], Viktor Fedun[2], and Olga Kryakunova[3]

[1]Scientific Space Systems Laboratory, Institute of Space Techniques and Technologies, 34, Kislovodskaya st., Almaty, Kazakhstan
[2]Plasma Dynamics Group, Department of Automatic Control and Systems Engineering, University of Sheffield, Mappin 6 Str., Sheffield, S1 3JD, United Kingdom
[3]Laboratory of Space Weather Diagnostic and Forecasting, Institute of Ionosphere, 117, Kamenskoe plato, Almaty, Kazakhstan

**Correspondence:** Anatoliy Lozbin (lozbin.a@istt.kz)

**Abstract.** In the Institute of Space Techniques and Technologies of the National Center of Space Research and Technology (Almaty, Republic of Kazakhstan) the DIAS (Detection of Ionosphere Anomalies Software) was developed and used for scientific research. The software was designed for ionosphere anomalies detection, identifying and analyzing from satellite spectral and wave data from scientific payload installed on the DEMETER spacecraft. The main task of DIAS Software is to provide
the researcher with a convenient tool for detection and identifying of the sources of electromagnetic radiation, disturbances of the ionic and electronic component of the ionosphere, and other ionosphere parameters from satellite data. Using this Software, a complex research of the state of the medium-latitude ionosphere during a geomagnetic storm on January 10, 2010 was done. Processing and analysis of the electric and magnetic components of the field in ULF, ELF and VLF band is carried out; as well as temperature, velocity and density of ionic and electronic plasma components and fluxes of energetic electrons at satellite
altitude during a storm.

## 1 Introduction

One of the most informative methods of ionosphere research is the method of measuring its parameters from the spacecraft. This method has a number of advantages and disadvantages. The main advantage of satellite methods of ionosphere research is the possibility of measurements directly inside of studied object. Using a set of measuring instruments on a satellite, a number
of ionosphere parameters can be obtained to study its connection with the atmosphere, lithosphere, solar activity, etc. If the satellite at the solar-synchronous orbit (altitude of 600-700 km), it makes about 15 orbit passes around the Earth per day, which provides the opportunity to explore the ionosphere over almost the entire Earth surface. However, this is also a problem, since the possibility of monitoring of short-term disturbances over certain regions of the Earth surface is practically excluded.

Man-made Earth's satellites (one or constellation) during active period, depending on the payload composition, accumulate
and transmit to ground a huge amount of data in its different form: raw (raw) data; partially processed data (waveform, spec-





trograms), etc. The amount of this data can reach tens of Gigabytes. Satellite scientific mission developers are often providing scientists with data in a limited form, for example, only time, coordinates and parameter value. Therefore, further research requires a great deal of time for scientific data processing by various software, especially if it concerns complex research.

The one of the interesting satellite projects for the study of the ionosphere and ionosphere-lithosphere coupling was the
French satellite DEMETER (Cussac et al., 2006), which active period was more than 6 years (2004-2010). In the of Scientific Space Systems Laboratory of the Institute of Space Techniques and Technologies (ISTT) of the National Center for Space Research and Technologies (Almaty, Republic of Kazakhstan) within the framework of the project of creation a Kazakhstan Satellite System for Scientific Purposes (Moldabekov et al., 2011), the development of a software for complex analysis of ionosphere parameters from satellite data was initiated and subsequently adapted for DEMETER data analysis.

Thus, the developed DIAS Software and data from scientific instruments on-board DEMETER spacecraft allow to provide a complex research of the various disturbances in the ionosphere and correlate them with natural and human-induced phenomena.

Undoubtedly, that magnetic storms, as a powerful change in the Earth's magnetic field due to the ejection of a huge number of high-speed streams (solar wind, shock wave, charged particles from space) during an increase in solar activity, somehow affect many parameters of the ionosphere. In this paper the state of the medium-latitude ionosphere according to its various
parameters from DEMETER Satellite and processed using the DIAS Software during a geomagnetic storm on January 20, 2010, are considered.

In Section 2 we review the DEMETER satellite and its scientific payload. In Section 3 we develop a new Software for complex analysis of ionosphere anomalies based on input data from DEMETER scientific instruments. The example of DIAS Software implementation for scientific research of ionosphere parameters during geomagnetic storm at January, 20, 2010 are
given in Section 4 and main results are outlined in Section 5.

## 2   DEMETER Satellite

DEMETER (Detection of Electro-Magnetic Emissions Transmitted from Earthquake Regions) is devoted to the investigation of the Earth ionosphere disturbances due to seismic and volcanic activities.

Of particular interest to DEMETER is that it was the first satellite to provide a complex research on ionospheric disturbances
associated with natural and artificial events on Earth.

The satellite weight was 130 kg and it was launched on June 29, 2004 from the Baikonur launch site into a solar-synchronous orbit of the Earth with a height of 710 km and an inclination of $98.3°$. According to the mission program the scientific on-board experiments was stopped on December 9, 2010.

The science payload is composed of five instruments:

1. ICE, three electric sensors from DC up to 3.5 MHz (Berthelier et al., 2006);

2. IMSC, three magnetic sensors from a few Hz up to 18 kHz (Parrot et al., 2006);

3. IAP, an ion analyzer (Berthelier et al., 2006);





4. IDP, an energetic particle detector (Sauvaud et al., 2006);

5. ISL, a Langmuir probe (Merikallio, 2006).

Data from scientific instruments onboard DEMETER spacecraft allow to provide a complex research of the various disturbances in the ionosphere and correlate them with natural and anthropogenic phenomena. ICE instrument is designed to

measure electrical disturbances in the ionosphere in ULF (Ultra Low Frequencies) band (0-20 Hz), ELF (Extremely Low Frequencies) band (0-1250 Hz), VLF (Very Low Frequencies) band (0-40 kHz) and HF (High Frequencies) band (0-3.3 MHz). IMSC magnetometer is designed to measure magnetic field disturbances in the ionosphere. The instrument consists of 3 orthogonal magnetic antennas connected to the preamplifier with a shielded cable 80 cm long. At the output of the IMSC device, the researcher is provided with magnetic data in ELF band (0-1250 Hz) and VLF band (0-40 kHz). The IAP ion analyzer is

designed to measure the composition of $H^+$, $He^+$ and $O^+$ ions, their density, temperature and velocity. The charged particle detector IDP is designed to detect weak flows of electrons in the range from 70 to 1000 keV. The output of the IDP for seismic regions of the Earth is the energy spectrum of electrons ($cm^2/s/ster/keV$). For the rest of the Earth surface, the electron fluxes for 3 energy ranges (90.7-526.8 keV; 526.8-971.8 keV and 971.8-2342.4 keV) are provided. The ISL Langmuir probe is designed to measure the density of electrons and their temperature.

## 3   DIAS Software

The DEMETER project team developed a different modules and software to extract and analyze data from output files, but all were not very convenient for the researcher. Therefore, it became necessary to create a universal tool that is simple and accessible to any user.The main task of the DIAS Software is to provide the researcher with a convenient tool for satellite data processing and analyses for detection and identifying the sources of various disturbances in the ionosphere.

The main functions of the system are:

– visualization of all types of scientific data from DEMETER spacecraft;

– data mapping on different types of maps;

– data multiupload (simultaneous upload a large amount of data);

– constructing of a spectrogram from a time series data row;

– possibility of calculating the signal-to-noise ratio for studying the effect of radio transmitters on the ionosphere.

### 3.1   Input data

DIAS Software uses data files with different APIDs (Application Process IDentifier - is DEMETER project files identifiers):

– waveform of three components of the electric field in the ULF band (APID 1129);



- waveform of three components of the electric field in the ELF band (APID 1130);

- waveform of one component of the electric field in the VLF band (APID 1131);

- spectrum of electric field in the VLF band (APID 1132);

- waveform of one component of electric field in HF band (APID 1133);

- spectrum of the electric field in the HF band (APID 1134);

- waveform of three magnetic field components in the ELF band (APID 1135);

- waveform of one component of the magnetic field in the VLF band (APID 1136);

- magnetic field spectrum in the VLF band (APID 1137);

- ions parameters (density, temperature and velocity) in "Burst" mode (APID 1139);

- ions parameters (density, temperature and velocity) in "Survey" mode (APID 1140);

- spectrum of energetic electrons in "Burst" mode (APID 1141);

- spectrum of energetic electrons in "Survey" mode (APID 1142);

- parameters of electrons (density, temperature and potential) in "Burst" mode (APID 1143);

- ion parameters (density, temperature and potential) in "Survey" mode (APID 1144).

## 3.2  Data Series Operation Mode

In data series operation mode the graph is plotted as shown in Figure 1. For multi-component measurements, all graphs are displayed in the same field in different colors with the ability to select the row of interest.

The main window consists of 3 parts:

- top main menu;

- right control panel;

- the graphics display area.

The top main menu of the Software allow the researcher work with data files, selected signals and images. The user can open, print or preview the data. It's possible to open one or several data files, print the resulting chart from the graphic display area, and the preview the image before printing or saving. When you click the "Signals" button, a row action window appears.

Researcher can perform the following actions on data series:

- choose to display or hide a series of data in the chart;





**Figure 1.** The screenshot of main window of the DIAS Software with an example of graph of electric field in VLF band

- to save the necessary data row in the "Time (DDMMYYHHMMSSMLS) — Value" format (MLS here is milliseconds, and the quantity of letters corresponds to the number of signs of date);

- to build an autospectogram for the selected data row using the fast Fourier transform algorithm;

- to perform high-frequency (HF) or low-frequency (LF) row smoothing;

5     – to save a data row as a *.wav audio file.

LF smoothing is performed as follows: for a given window there is an average value of the parameter, which is entered in the first point of the new row. The window is shifted by 1 point and the action is repeated. Then, its LF component is subtracted from the initial data row. The HF component is the difference between the original data row and its LF component.

Also, for the selected time series, the software builds a spectrogram - the spectrum module in the sliding windows is calcu-
10  lated. The length of the spectrogram is equal to the length of the original data row. Each column of the histogram is a spectrum





that has a length equal to half the length of the window. When the cursor moves along the spectrogram field, time (horizontally), frequency (vertically) and spectrum amplitude value are shown. The right control panel in time series data operation mode is intended for control and operations with values loaded into the software.

The top two fields "Y max." and "Y min." are used to scale along the $Y$ axis.

The two following fields, "Time, from" and "Time, to", are used to scale along the $X$ axis.

The "Resolution" feature allows to set arbitrary values for the chart size as desired by the user.

The "Curve" information field displays the service information about the chart: the number of points, grid spacing along the $X$ and $Y$ axes.

The small map in the "Data at cursor" field displays the position of the satellite for the loaded data on the world map, and it
also acts as a link for map window opening.

"Details" section give a detailed information for each graph point. The composition of parameters varies depending on the type of data.

The data service information is contained in each data file and consists of 4 blocks:

- Block 1: General Header;

- Block 2: Orbital and Geomagnetic Parameters;

- Block 3: Attitude Parameters;

- Block 4: Measured values.

### 3.3  Spectral Data Operation Mode

Some of the data files (APID 1132, APID 1134, APID 1137 and APID 1141) contain spectral information which displayed as
spectrgram in the central window of the Software.The examples of such spectral data visualisation will follows.

Same as in time series rows mode, the Software window in the spectral data mode is also divided into 3 parts: the upper main menu; right control panel and graphic display area.

In the graphic area, a spectrogram is displayed, where time and geographical coordinates of the satellite position are given for electromagnetic data on the $X$ axis, for the $Y$ axis - frequency in Hertz, and on the $Z$ axis in the color scale the measured values
are presented in the log scale $log(\mu V^2/m^2/Hz)$ for electrical measurements or $log(nT^2/Hz)$ - for magnetic. For the spectrum of energetic electrons along the $X$ axis, time and geographical coordinates of the satellite position are also given, for the $Y$ axis, the electron energy (in keV), and for the $Z$ axis measured values are given in the color scale in units $part/cm^2/ster/keV$.

The following information can be obtained from spectrogram:

- amplitude of the spectrum at a certain frequency;

- spectral density;

- signal-to-noise ratio for disturbances at radio transmitter operating frequencies;





– search for "whistles" caused by lightning discharges.

There is possible to get a graph of values at a certain frequency (Curve on frequency, see Figure 2). Similarly, the "Spectral density" option allows to display the spectral density for a given frequency range as a graph. The "Signal-to-Noise Ratio"

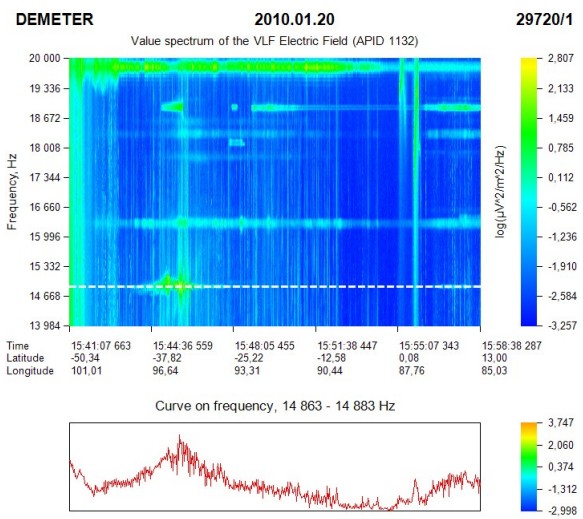

**Figure 2.** Spectrum of the electric component of the electro-magnetic field and amplitude at the operational frequency 14,881 kHz of the VLF transmitters of the Alfa (RSDN-20) radio system.

function provides a diagram for a given ratio for a particular transmitter. The "Base Frequency" field indicates the frequency of
5 the transmitter, and the "Frequency df" window indicates the frequency range over which the radiation is propagated from the base frequency. The algorithm described in article (Molchanov et al., 2006) was used for this option. Here, the signal-to-noise ratio was determined by the formula:

$$SNR = 2A(F_0)/[A(F_+) + A(F_-)]$$

where, $A(F_0)$ is the spectral density amplitude at transmitter operating frequency $F_0$, $A(F_+)$ and $A(F_-)$ are amplitudes of
10 spectral density of disturbance width higher and lower than base frequency of transmitter respectively.

The result of such analysis may be a pattern of electric field disturbance in the southern and northern magneto-conjugate regions above the VLF transmitter NWC 19,8 kHz (see Figure 3). The DIAS Software has the possibility of displaying the amplitude of the analysed parameter in color along the trajectory of the satellite, which makes it possible the visual evaluation of the changes in ionosphere over the studied territory.

The "Whistlers Searching" function allows user to research the disturbance of the ionosphere caused by lightning discharges using vertical structures on a spectrogram (thin vertical lines - "whistlers"). The DIAS analyse the spectrogram of the electric and magnetic field in the frequency band from 2 to 10 kHz and find whistles corresponding to the coefficients settled by user. The coefficients control the spectral width of the whistle search, i.e. its sensitivity. The result of the analysis is given at the bottom of the spectrogram, as shown in Figure 4.



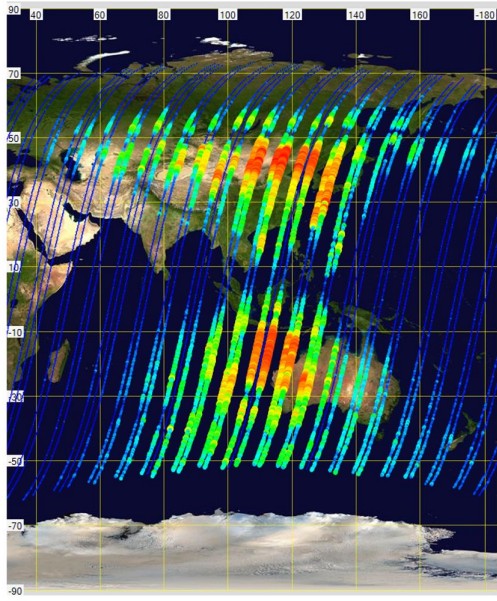

**Figure 3.** The electric field disturbance in the southern and northern magneto-conjugate regions above the VLF transmitter NWC 19,8 kHz.

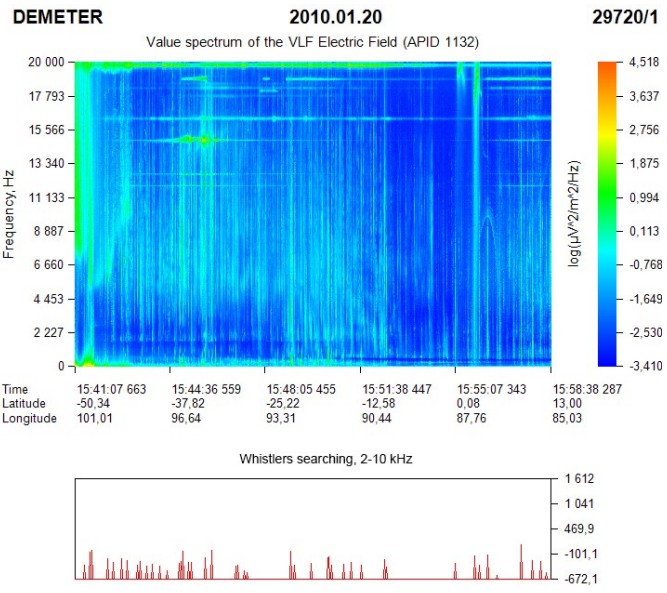

**Figure 4.** Spectrogram of the electric field in the VLF band with the result of whistlers searching with coefficient 3.

Also, it's possible to export the amplitude of the spectrogram for the selected time point to the *.csv format.

To study the low-energy electrons precipitation at the height of low-orbital satellites, in addition to spectrograms displaying, it is also possible to obtain a graph for the selected energy, energy range (spectral density) and for a time point. An example of





such a spectrogram is shown in Figure 5. Here, below the spectrogram window, a graph is shown for electrons with an energy of 108.5 keV for the DEMETER half-orbit No.29720/1, 20.01.2010.

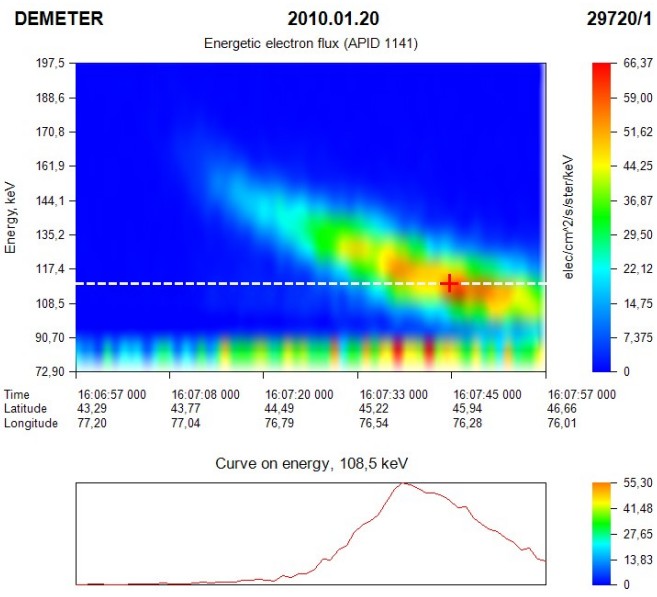

**Figure 5.** The energetic electrons spectrogram and graph for electrons with an energy of 108.5 keV for the ascending half-orbit of spacecraft DEMETER No.29720/1, 20.01.2010.

### 3.4 Data mapping

One more important function of the DIAS Software it is data mapping - showing data place and value on different map types.

The software supports 6 types of maps: 4 offline maps (physical, topographic, tectonic and contour) and 2 online maps (Google Maps (satellite, political and terrain) and Bing Maps.

The Software allow to operate of the different visual effects of the satellite track and user defined objects on the maps. User can see the satellite track and show or hide the visualisation of the south and north magnetically conjugate points at the altitude 110 km.

Also, the researcher can display on the map along the orbit track the values of the graph at frequency, spectral density and signal-to-noise ratio. The amplitude is displayed as circles with different diameter and color depending on its value - from a small point of blue color with minimum values to a circle of larger size of red color at maximum (Figure 3).

### 3.5 Data multiupload

Another advantage of DIAS is the data multiupload, i.e. the simultaneous loading and processing of a large amount of similar

type data. Some scientific tasks require the analysis of data over a geographical area of the Earth (the impact of LF transmitters





on the ionosphere, the study of the earthquakes precursors, etc.), and since the satellite pass through the same territory about once per 3 days, scientists face the task of choosing the right half-orbits and a complex analysis of data from these half-orbits.

The DIAS allows you to simultaneously load data with the same APID. It is allowed to load data together with APID 1143 and 1144, as well as APID 1139 and 1140, since this is the same type data, but with different measurement frequency.

## 5  4   Analysis of the middle-latitude ionosphere parameters during the geomagnetic storm at January, 20, 2010

On January 20, 2010, various geomagnetic observatories of the world detected a geomagnetic storm (see Figure 6). The maximum of Kp index was on 15-18 hours by UTC. According to the "Alma-Ata" geomagnetic observatory, at the same time, magnetic field disturbances were observed in all 3 directions (see Figure 7).

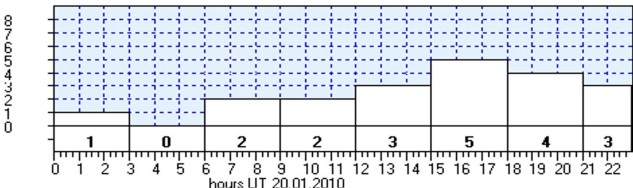

**Figure 6.** Values of Kp index on 20.01.2010.

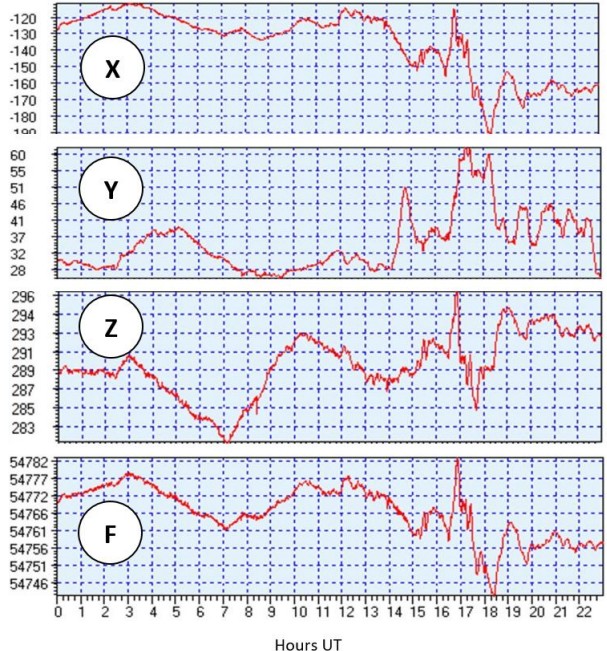

**Figure 7.** The intensity of the geomagnetic field at the "Alma-Ata" Observatory on 20.01.2010.





At the same time, according to the website www.spaceweatherlive.com there was an increase in the parameters of the solar wind and the intensity of the interplanetary magnetic field (Figure 8). Here we see that the increase in the speed of the solar wind began from about 08:00 UTC. A sharp jump occurred at about 17:00 UTC, and the maximum occurred at 23:40 UTC. In general, the increase in the speed of the solar wind compared to 08:00 UTC was $\sim 80$ %.

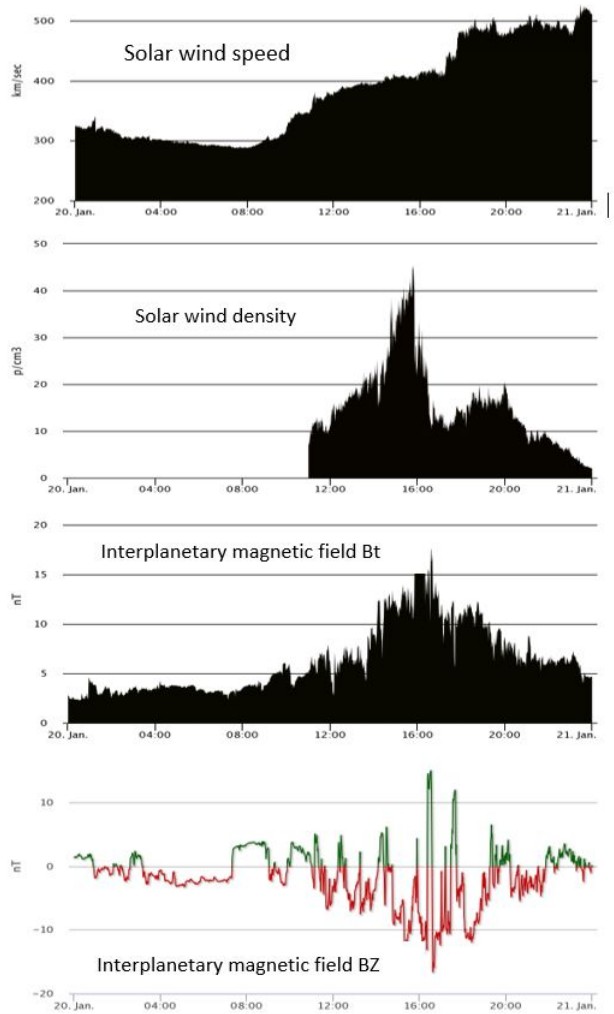

**Figure 8.** Parameters of solar wind and interplanetary magnetic field on 20.01.2010.

Also, at 15:50 UTC, a maximum solar wind density up to 45 $part/cm^3$ was observed. From 08:00 to 23:00 UTC, the intensity of the interplanetary magnetic field increased from 3 to 18 nT. According to the same site, there was no increase in the level of solar protons with energy of more than 10 MeV on January 20, 2010.

During the maximum of these events ($\sim 16 : 00$ UTC), the ascending orbit of the DEMETER spacecraft passed over the territory of Kazakhstan. The initial half-orbit time is 15:40 UTC, and the final - 16:15 UTC.





No obvious disturbances were observed in the ULF electric field band (0-20 Hz). Analysis of the electric field in the ELF band at satellite altitude showed the presence of low-frequency electrostatic vibrations on the spectrum (horizontal structures in the left part of the spectrum in Figure 9). These are bursts of electrical oscillations at 100-400 Hz frequencies, which are usually observed on the satellite in the zone of aurora polaris at various altitudes. Most often, these bursts are explained either

5    by the crossing of small-scale spatial charge structures by sattelite, or by the current generation of ion-sound or ion-cyclotron oscillations (Molchanov, 1985). However, in our case, these disturbances are observed at medium latitudes ($32° - 47°$ N), but it is quite difficult to associate them with a geomagnetic storm, since for other half-orbits, under calm geomagnetic conditions, such fluctuations were also observed.

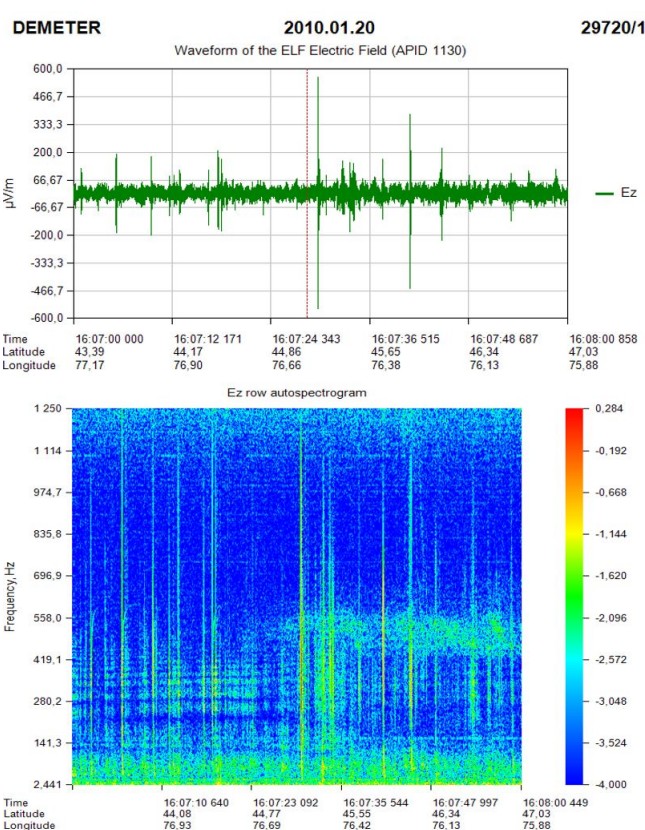

**Figure 9.** Graph and spectrum of electric field strength in the ELF band for the ascending half-orbit No.29720.

In the right part of the spectrogram in Figure 9, plasmospheric hisses are apparently observed in the frequency band $\sim$

10    $450 - 550$ Hz. These radiations occur quite often and are dispersed throughout the plasmosphere with minor variations in intensity. In our case, apparently, these noises were amplified just by electrons precipitation. The vertical structures on this spectrogram are electronic whistles caused by lightning discharges in the atmosphere.

Data from the IMSC magnetometer is quite "noisy" and it is not possible to analyze fine structures in this case.



According to data from the IAP instrument (Figure 10), we can see that there was a increase in the temperature and velocity of ions from 16:03 to 16:08 UTC with a maximum of 16:05:40, over the Central Tien Shan area $\sim 41-42°$ N (northern magneto-conjugate point). Also, there was a significant decrease in the concentration of oxygen ions and its alignment over the area of the Gobi desert at the level of 30-40 $part/cm^3$. In the helium and hydrogen ions density, minor variations and a general increasing tendency to north in latitude were observed.

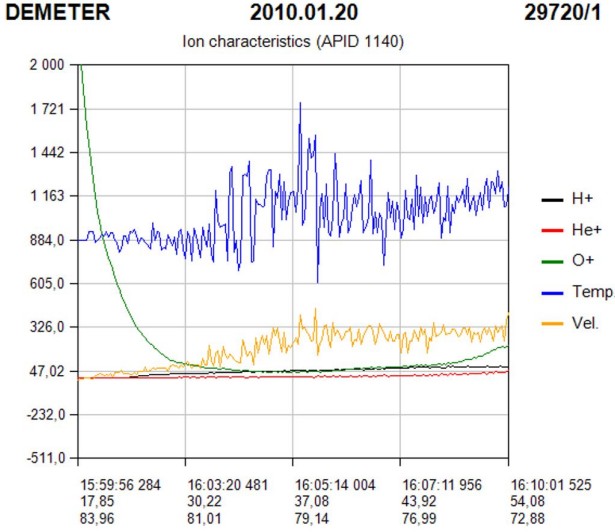

**Figure 10.** Temperature, velocity and density and of ions for half-orbit No.29720.

The spectrogram of the IDP instrument (Figure 5) shows that from 16:07:15 to 16:08:33 electrons with an energy of 160 keV from the Earth's internal radiation belt are precipitated.

The behavior of energetic electrons trapped in the Earth's Van Allen radiation belts has been extensively studied, through both experimental and theoretical techniques. During quiet period, energetic radiation belt electrons are distributed into two belts divided by the electron slot at $L \sim 2.5$, near which there is relatively low energetic electron flux.

It is well known that large-scale injections of energetic particles into the outer radiation belts are associated with geomagnetic storms which can increase trapped electron population up to 105 times (Li and Temerin, 2001). In most cases, these injections do not penetrate into the inner radiation belt. Only in case of biggest storms, the inner belt gain a new population of energetic electrons (e.g. Baker et al., 2004).

Observations by the DEMETER near the powerful VLF transmitter NWC have shown that this transmitter causes heating of electrons and ions in the ionosphere at the satellite altitude (700 km), affecting a $\sim 500,000 km^2$ area (Parrot et al., 2007). There was shown increases in energetic electrons in the range 91–527 keV caused to NWC. Further experiments based on DEMETER wave and particle data show the significance of NWC upon electrons in the inner radiation belt and enhancements in the $\sim 100˘600 keV$ drift-loss cone electron fluxes at low L values are linked to NWC operation and to ionospheric absorption





(Sauvaud et al., 2008). Such enhancements, termed "wisps" and looking like on Figure 15, are only detected eastward of the transmitter location, as expected from the electron drift motion, and at energies that are consistent with first-order equatorial cyclotron resonance between the NWC transmissions and electrons interacting in the vicinity of the magnetic equatorial plane (Gamble et al., 2008). These authors conclude that the NWC transmitter have a good position for potential influence upon inner
radiation belt > 100 keV electrons.

In our case, the region in which an increasing of energetic electrons has been observed is the northern magneto-conjugate region of the NWC transmitter and it may well be, that the observed electron precipitation are related to the operation of the NWC transmitter and the effects of a magnetic storm.

## 5   Conclusions

An original instrument was created – "DIAS" Software for detection, identification and analyzing of ionosphere anomalies from satellite spectrograms and time series rows data from instruments onboard DEMETER satellite. Using this Software, in the ISTT the research of ionosphere parameters variations caused by various anthropogenic and natural factors are provided.

A Certificate of the Republic of Kazakhstan on the entry of information into the state register of rights to objects protected by copyright No.11767 of 26.08.2020 was received for the developed Software.

The scientific data processing and visualization technologies used in the development of DIAS Software can be used in the creation of Software for other scientific space missions.

The application of the developed Software on the example of complex analysis of the state of the medium-width ionosphere according to its various parameters based on satellite data during a geomagnetic storm on January 20, 2010 is considered. According to these data, the presence of electrostatic vibrations in the frequency range of 100-400 Hz, which are quite difficult
to connect directly with the geomagnetic storm, was revealed in the ELF spectrum of electric field strength. Also, there was variations in the temperature and velocity of ions, however, they fall at lower latitudes and it is also quite difficult to connect them with the effects of a geomagnetic storm, since similar variations were observed for quite half-orbits. During the maximum of the geomagnetic storm, electrons with an energy of 160 keV from the Earth's internal radiation belt are precipitated.

*Author contributions.* AL proposed the idea and concept of the paper, made analytical calculations, wrote the initial version of the paper,
and took part in the revision of the paper. VF and OK contributed to the data analysis and paper writing. All the authors took part in the results analysis, developing the concept of the paper and writing the initial version of the paper. AL and VF contributed to the preparation of the final version of the paper.

*Competing interests.* The authors declare that they have no conflict of interest.





*Acknowledgements.* This research was performed as a part of doctoral project of PhD student of Kazakh National University named after al-Farabi, Deputy Head of Scientific Space Systems Laboratory of ISTT – A. Lozbin.





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
