# Peer review of "Complex analysis of the ionosphere variations during the geomagnetic storm at January, 20, 2010 performed by DIAS software and DEMETER satellite data"

_Annales Geophysicae, 2021_

## Referee Comment (RC1)

REVIEW
Authors: Anatoliy Lozbin, Viktor Fedun, and Olga Kryakunova
Title: «**Complex analysis of the middle-latitude ionosphere parameters during the geomagnetic storm at Jan, 20, 2010 based on the DEMETER satellite data analysed using DIAS Software**»

The paper presents software for effectively processing data from the DEMETER satellite. The processing aims at searching for the effects in geospace that are caused by different sources. The study is urgent since the DEMETER satellite has collected a large amount of data requiring further processing.

The software performance is illustrated by some results of data processing.

The paper layout is quite successful.

The manuscripts need some improvements.

(1) The paper is of an advertising character since it contains few physical results. The storm effects are actually absent. The January 20, 2010 storm is described in the literature. The authors should compare their results with the results obtained by others (see, e.g., the results obtained by the incoherent scatter technique [Domnin, I. F., Emelyanov, L. Ya., Pazura, S. A., Kharytonova, S. V., Chernogor, L. F. Dynamic processes in the ionosphere during the very moderate magnetic storm on 20-21 January 2010 (In Russian) // Space Science and Technology. 2011. Vol. 17, no. 4. Pp. 26–40].

The authors should have considered a strong storm.

(2) The authors allegedly discovered the effects arising from the particle precipitation during the storm. However, precipitations from the inner radiation belt can only occur during strong storms. [Baker, D. N., Kanekal, S. G., Li, X., Monk, S. P., Goldstein, J., and Burch, J. L.: An extreme distortion of the Van Allen belt arising from the 'Hallowe'en' solar storm in 2003, 432, 878–881, https://doi.org/10.1038/nature03116, 2004.].

(3) The authors assert (line 10-15) that magnetic storms affect ionospheric parameters. This approach seems outdated. Magnetic and ionospheric storms, like atmospheric and electrical storms, are components of a single process, namely, a geospace storm (see, e.g, Chernogor L. F., Garmash K. P., Guo Q., Zheng Y. Effects of the Strong Ionospheric Storm of August 26, 2018: Results of Multipath Radiophysical Monitoring / L. F. Chernogor, K. P. Garmash, Q. Guo, Y. Zheng // Geomagnetism and Aeronomy. – 2021. – Vol. 61, No. 1. – Pp. 73–91; Chernogor L. F., Garmash K. P., Guo Q., Luo Y., Rozumenko V. T., Zheng Y. Ionospheric storm effects over the People's Republic of China on 14 May 2019: Results from multipath multi-frequency oblique radio sounding / L. F. Chernogor, K. P. Garmash, Q. Guo, Y. Luo, V. T. Rozumenko, Y. Zheng // Advances in Space Research. – 2020. – Vol. 66, Is. 2. – Pp. 226–242; Luo Y., Chernogor L. F., Garmash K. P., Guo Q., Rozumenko V. T., Zheng Y. Dynamic processes in the magnetic field and in the ionosphere during the 30 August–2 September, 2019 geospace storm. Annales Geophysicae. https://doi.org/10.5194/angeo-2020-57 .

(4) The authors mistakenly state that "… effect of radio transmitters on the ionosphere" (line 25). The ionosphere is actually affected by the radio emissions from the transmitter.

(5) It is necessary to expand the figure captions, to make them more informative.

(6) $K_{p\mathrm{max}}$ should be specified.

Recommendation: Return to authors for minor revisions

Sincerely,
Reviewer.

---

## Referee Comment (RC2)

Comments on the manuscript "Complex analysis of the middle-latitude ionosphere parameters during the geomagnetic storm at Jan, 20, 2010 based on the DEMETER satellite data analysed using DIAS Software" by A. Lozbin, V. Fedun and O. Kryakunova.

The paper describes the software that may be useful for DEMETER satellite data processing and presents the case study using these data. This is an interesting article that I would recommend for publication after revision. The majority of my comments are minor.

**General comments.**

(1) My main recommendation is to state clearly whether the authors describe the software features or present the analysis of the particular magnetic storm effects on the ionosphere. Now it is confusing: the first part of the article seems to be a program manual and the second - a case study with a missing scientific focus.
I recommend to state clearly the aim and the tasks of the study (p.2 line 14). There is a lot of work done, which is much appreciated. The authors just need to put in "frame".

(2) I have a doubt about the "DIAS" acronym. Up to my knowledge, it is widely used for European Digital Upper Atmosphere Server (DIAS) initiated in 2004. Please see the works of Belehaki et al. Probably, some clarification is needed here.

(3) The Acknowledgement of work of developers of the DEMETER satellite equipment is missed.

(4) Where the developed DIAS software may be accessed?

In case that the authors would like to present a full research study:

(5) p.11 line 1 and further: Why do you discuss the interplanetary parameter variations? Is it important for your analysis of the ionosphere state change? I recommend only a brief description - a couple of sentences with citing the appropriate works.

(6) What exactly can be concluded on the changes in the ionosphere by your analysis? Over what area?

(7) I would expect some references to the papers that already discussed the considered magnetic storm. What new was found?

(8) p.1 line 18. I would add that due to the fact that the satellite passes over the different parts of the Earth, it is impossible to take into account the diurnal variation of ionospheric parameters over some particular point of observation. The last is rather important when searching for the irregular parameter behaviour. I recommend discussing this in the text.

**Minor comments.**

I suggest replacing "*disturbances in the ionosphere*" with "ionospheric variations" throughout the text.

Please replace *UTC* with "UT" throughout the manuscript.

Please note, that first the term should me mentioned and then its acronym should be introduced, not otherwise. For instance, p.3 line 5: *ULF (Ultra Low Frequencies)* → Ultra Low Frequencies (ULF). Please revise carefully throughout the text.

Title: Please replace *Jan* with "January" and *Based on the* with "by".
In general, the title is long and confusing. I recommend changing it according to the aim of the paper.

1-12: I am not sure that the *measurements* (these or that) are a *method*. These are two different concepts. I suggest calling them experiment/ obtaining data/ satellite measurements, but not a method.

1-19:
*Man-made* → artificial
Eliminate *during active period*
Eliminate *composition,*

1-20: *raw (raw)* - Eliminate repetition.

2-21: *Providing* → provide

2-1: Why limited? What else is needed except for the time, coordinates and value?

2-5: *In the of Scientific….*
The sentence is too large and difficult to follow. Please separate it into several sentences.

2-12:
*Undoubtedly, that* → It is known that
Eliminate *that*

2-22: *is devoted* → was

2-25: *Events* → hazards

2-29: *science payload* → scientific payload

2-30: Five instruments are mentioned by their acronyms. The acronyms must be introduced.

3-3: *Data from scientific in…*
Please eliminate this sentence as it repeats the said above.

3-10 and further: I am not sure I follow the idea. It is stated that the detector works in two regimes: for seismic regions and for the rest of the Earth surface. Is it correct? The authors probably meant that the detector was capable of measuring different ranges of energies. Please explain clearly.

3-25: *possibility of calculating the signal-to-noise ratio* → signal-to-noise ratio calculation

Subsection 3.1: Please indicate how the discussed files can be accessed. Downloaded from some web-page?

4-23: *Allow* → allows

5-11: *Also,* → In addition,

7-1: *There is possible to get a graph of* → It is possible to plot

7-11: *result of such analysis may be a pattern* – I am not sure I understand the meaning of this sentence. Please rephrase.

7-12: *NWC transmitter* - Please introduce the acronym and provide the details on the transmitter (transmitter network?). Where is the receiver?
14-2: eastward of the transmitter location?

9-4: *One more important* → Another

9-5: What do you mean by *physical map*?

10-2: I recommend explaining the meaning of the *right half-orbits* (what right or left half means) and what do you imply by choosing them.
The same for: 11-9.

10-5: *Storm at* → storm on

10-7:
*maximum of Kp index was on 15-18 hours by UTC.* → Kp reached its maximum value between 15 and 18 UT.
*directions* → magnetic field components (?)

11-1: If the authors use the data/information from some internet source, they should clearly state why and for what purpose. None web-page should be cited without a proper explanation about whose page is it and why the authors use its data/information.

12-4: On → by

12-4: *zone of aurora polaris at various altitudes*
Auroral zone? What latitudes do you mean?

12-5: *charge structures* - ?
May be replace this with "disturbances" or "irregularities"?

12-8: *half orbits* - Please explain where it is exactly.

12-9: *right* → lower ?

12-11:
The word *apparently* is repeated several times.
Eliminate *just*.

Figure 9: Please indicate with arrows the whistlers and the plasmospheric hisses in the figure

13-3: *magneto-conjugate point*
Magneto-conjugate point of what?

13-6 and further:
It is important over which latitudes the electron precipitations of these or that energies are registered. Please be more specific about what the observed precipitations mean for the ionospheric effects (geophysically).

14-17: width → latitude

---

## Author Comment (AC1)

**Reply to the Referee**
We are grateful to the referee for his/her useful comments and appreciate very much his/her efforts in improving our paper. The authors note that the referee has carefully and critically evaluated our paper. We have provided answers to all the issues raised in this report below. In addition, our revisions within the manuscript are highlighted in black colour to assist the reviewer.

**Referee report**

Authors: Anatoliy Lozbin, Viktor Fedun, and Olga Kryakunova
Title: **«Complex analysis of the middle-latitude ionosphere parameters**
**during the geomagnetic storm at Jan, 20, 2010 based on the**
**DEMETER satellite data analysed using DIAS Software»**
The paper presents software for effectively processing data from the DEMETER satellite. The processing aims at searching for the effects in geospace that are caused by different sources. The study is urgent since the DEMETER satellite has collected a large amount of data requiring further processing. The software performance is illustrated by some results of data processing. The paper layout is quite successful. The manuscripts need some improvements.

**Referee comment**
(1) The paper is of an advertising character since it contains few physical results. The storm effects are actually absent. The January 20, 2010 storm is described in the literature. The authors should compare their results with the results obtained by others (see, e.g., the results obtained by the incoherent scatter technique [Domnin, I. F., Emelyanov, L. Ya., Pazura, S. A., Kharytonova, S. V., Chernogor, L. F. Dynamic processes in the ionosphere during the very moderate magnetic storm on 20-21 January 2010 (In Russian) // Space Science and Technology. 2011. Vol. 17, no. 4. Pp. 26–40].
The authors should have considered a strong storm.

**Our reply**

The main idea of this paper is to tell about an absolutely new instrument for researcher, using, maybe, not the best example. But, even in this case the complex analysis of parameters of the ionosphere was performed.

**Referee comment**
(2) The authors allegedly discovered the effects arising from the particle precipitation during the storm. However, precipitations from the inner radiation belt can only occur during strong storms. [Baker, D. N., Kanekal, S. G., Li, X., Monk, S. P., Goldstein, J., and Burch, J. L.: An extreme distortion of the Van Allen belt arising from the 'Hallowe'en' solar storm in 2003, 432, 878–881, https://doi.org/10.1038/nature03116, 2004.].

**Our reply**

As you can see on Spectrogram 5, the presence of energetic electrons with energy 100-150 keV is no doubt. Truly, electrons precipitation can be caused only by strong storms and this is noticed in paper. Also, it is considered that this precipitation can be caused by VLF transmitter activity (in our case it is the north magneto-conjugate zone of NWC transmitter). Nevertheless, such

precipitation is very rare (just several per year), so, I think, there should be additional conditions for that effect.

**Referee comment**
(3) The authors assert (line 10-15) that magnetic storms affect ionospheric parameters. This approach seems outdated. Magnetic and ionospheric storms, like atmospheric and electrical storms, are components of a single process, namely, a geospace storm (see, e.g, Chernogor L. F., Garmash K. P., Guo Q., Zheng Y. Effects of the Strong Ionospheric Storm of August 26, 2018: Results of Multipath Radiophysical Monitoring / L. F. Chernogor, K. P. Garmash, Q. Guo, Y. Zheng // Geomagnetism and Aeronomy. – 2021. – Vol. 61, No. 1. – Pp. 73–91; Chernogor L. F., Garmash K. P., Guo Q., Luo Y., Rozumenko V. T., Zheng Y. Ionospheric storm effects over the People's Republic of China on 14 May 2019: Results from multipath multi-frequency oblique radio sounding / L. F. Chernogor, K. P. Garmash, Q. Guo, Y. Luo, V. T. Rozumenko, Y. Zheng // Advances in Space Research. – 2020. – Vol. 66, Is. 2. – Pp. 226–242; Luo Y., Chernogor L. F., Garmash K. P., Guo Q., Rozumenko V. T., Zheng Y. Dynamic processes in the magnetic field and in the ionosphere during the 30 August–2 September, 2019 geospace storm. Annales Geophysicae. https://doi.org/10.5194/angeo-2020-57 .

**Our reply**

Here you are right and this sentence is changed taking into account your remarks.

**Referee comment**
(4) The authors mistakenly state that "… effect of radio transmitters on the ionosphere" (line 25). The ionosphere is actually affected by the radio emissions from the transmitter.

**Our reply**
Corrected

**Referee comment**
(5) It is necessary to expand the figure captions, to make them more informative.

**Our reply**
Done

**Referee comment**
(6) $Kp$max should be specified.

**Our reply**
Done

We hope that after these corrections the referee will find our MS suitable for publication in *Annales Geophysicae.*

On behalf of all the authors
Sincerely Yours

Anatoliy Lobzin

---

## Author Comment (AC2)

**Reply to the Referee**

We are grateful to the referee for his/her useful comments and appreciate very much his/her efforts in improving our paper. The authors note that the referee has carefully and critically evaluated our paper. We have provided answers to all the issues raised in this report below. In addition, our revisions within the manuscript are highlighted in black colour to assist the reviewer.

**Referee report**

Comments on the manuscript "Complex analysis of the middle-latitude ionosphere parameters during the geomagnetic storm at Jan, 20, 2010 based on the DEMETER satellite data analysed using DIAS Software" by A. Lozbin, V. Fedun and O. Kryakunova. The paper describes the software that may be useful for DEMETER satellite data processing and presents the case study using these data. This is an interesting article that I would recommend for publication after revision. The majority of my comments are minor.

**Referee comment**

(1) My main recommendation is to state clearly whether the authors describe the software features or present the analysis of the particular magnetic storm effects on the ionosphere. Now it is confusing: the first part of the article seems to be a program manual and the second - a case study with a missing scientific focus.
I recommend to state clearly the aim and the tasks of the study (p.2 line 14). There is a lot of work done, which is much appreciated. The authors just need to put in "frame".

**Our reply**

The main idea of this paper is to tell about an absolutely new instrument for researchers, using, maybe, not the best example. But, even in this case the complex analysis of parameters of the ionosphere was performed. The paper only with Software description will not look like a scientific paper. So, we decided to show how scientists can use this instrument for their research.

**Referee comment**

(2) I have a doubt about the "DIAS" acronym. Up to my knowledge, it is widely used for European Digital Upper Atmosphere Server (DIAS) initiated in 2004. Please see the works of Belehaki et al. Probably, some clarification is needed here.

**Our reply**

Sorry, but at the moment of Software development we don't hear anything concerning European Digital Upper Atmosphere Server (DIAS). So, at this moment it will be too hard to change the name of our Software. But, I think it is possible to remove the acronym "DIAS" from the paper.

**Referee comment**

(3) The Acknowledgement of work of developers of the DEMETER satellite equipment is missed.

**Our reply**

The acknowledgement of satellite developers is added, The links of scientific payload developers are present in the references.

**Referee comment**
(4) Where the developed DIAS software may be accessed?
In case that the authors would like to present a full research study:

**Our reply**
At this moment this Software is not available online, but anybody who wants to get it can send us a mail and we will be glad to send it personally.

**Referee comment**
(5) p.11 line 1 and further: Why do you discuss the interplanetary parameter variations? Is it important for your analysis of the ionosphere state change? I recommend only a brief description - a couple of sentences with citing the appropriate works.

**Our reply**
Agree. This part is removed.

**Referee comment**
(6) What exactly can be concluded on the changes in the ionosphere by your analysis? Over what area?

**Our reply**
During the maximum of the geomagnetic storm, electrons with an energy of 160 keV from the Earth's internal radiation belt are precipitated, but the reason (geomagnetic storm or radio transmitters or something else) of such even is not clear.

**Referee comment**
(7) I would expect some references to the papers that already discussed the considered magnetic storm. What new was found?

**Our reply**
This storm is not the biggest event, so there are not many papers about it.

**Referee comment**
(8) p.1 line 18. I would add that due to the fact that the satellite passes over the different parts of the Earth, it is impossible to take into account the diurnal variation of ionospheric parameters over some particular point of observation. The last is rather important when searching for the irregular parameter behaviour. I recommend discussing this in the text.

**Our reply**
Added

**Referee comment**
I suggest replacing "*disturbances in the ionosphere*" with "ionospheric variations" throughout the text.

Please replace *UTC* with "UT" throughout the manuscript.

Please note, that first the term should me mentioned and then its acronym should be introduced, not otherwise. For instance, p.3 line 5: *ULF (Ultra Low Frequencies)* Ultra Low Frequencies (ULF). Please revise carefully throughout the text.

Title: Please replace *Jan* with "January" and *Based on the* with "by".

In general, the title is long and confusing. I recommend changing it according to the aim of the paper.

**Our reply**
Done

**Referee comment**
1-12: I am not sure that the *measurements* (these or that) are a *method*. These are two different concepts. I suggest calling them experiment/ obtaining data/ satellite measurements, but not a method.

**Our reply**
Done

**Referee comment**
1-19: *Man-made* -> artificial
Eliminate *during active period*
Eliminate *composition,*

**Our reply**
Done

**Referee comment**
1-20: *raw (raw)* - Eliminate repetition.

**Our reply**
Done

**Referee comment**
2-21: *Providing* ->provide

**Our reply**
Done

**Referee comment**
2-1: Why limited? What else is needed except for the time, coordinates and value?

**Our reply**
Here we means that sometimes scientists are not a programmers and data processing takes a time. However, I think it will be better to replace this ny the word "raw" .

**Referee comment**

2-5: *In the of Scientific….*
The sentence is too large and difficult to follow. Please separate it into several sentences.

**Our reply**
Done

**Referee comment**
2-12: *Undoubtedly, that->* It is known that
Eliminate *that*

**Our reply**
Done

**Referee comment**
2-22: *is devoted ->*was

**Our reply**
Done

**Referee comment**
2-25: *Events ->*hazards

**Our reply**
Done

**Referee comment**
2-29: *science payload ->*scientific payload

**Our reply**
Done

**Referee comment**
2-30: Five instruments are mentioned by their acronyms. The acronyms must be introduced.

**Our reply**
Done

**Referee comment**
3-3: *Data from scientific in…*
Please eliminate this sentence as it repeats the said above.

**Our reply**
Done

**Referee comment**

3-10 and further: I am not sure I follow the idea. It is stated that the detector works in two regimes: for seismic regions and for the rest of the Earth surface. Is it correct? The authors probably meant that the detector was capable of measuring different ranges of energies. Please explain clearly.

**Our reply**

The measurement frequency of the instruments are not changed. But due to the amount of data and transmission rate limit the numerical data can be obtained only under the seismic regions (for some instruments). For the rest of the world only the spectral data are available (performed by FFT).

**Referee comment**

3-25: *possibility of calculating the signal-to-noise ratio* ->signal-to-noise ratio calculation
Subsection 3.1: Please indicate how the discussed files can be accessed. Downloaded from some web-page?

**Our reply**

Done. Webpage link is added.

**Referee comment**

4-23: *Allow* ->allows

**Our reply**

Done

**Referee comment**

5-11: *Also,* ->In addition,

**Our reply**

Done

**Referee comment**

7-1: *There is possible to get a graph of* ->It is possible to plot

**Our reply**

Done

**Referee comment**

7-11: *result of such analysis may be a pattern* – I am not sure I understand the meaning of this sentence. Please rephrase.

**Our reply**

Done

**Referee comment**

7-12: *NWC transmitter* - Please introduce the acronym and provide the details on the transmitter (transmitter network?). Where is the receiver?

**Our reply**
Done

**Referee comment**
14-2: eastward of the transmitter location?

**Our reply**
Corrected

**Referee comment**
9-4: *One more important->* Another

**Our reply**
Done

**Referee comment**
9-5: What do you mean by *physical map*?

**Our reply**
It means geographical map. Corrected.

**Referee comment**
10-2: I recommend explaining the meaning of the *right half-orbits* (what right or left half means) and what do you imply by choosing them.
The same for: 11-9.

**Our reply**
It means half-orbit under the region of interest. Corrected.

**Referee comment**
10-5: *Storm at ->* storm on
10-7:
*maximum of Kp index was on 15-18 hours by UTC*. ->Kp reached its maximum value between 15 and 18 UT.
*directions->* magnetic field components (?)

**Our reply**
Done

**Referee comment**
11-1: If the authors use the data/information from some internet source, they should clearly state why and for what purpose. None web-page should be cited without a proper explanation about whose page is it and why the authors use its data/information.

**Our reply**

Clearly

**Referee comment**

12-4: On ->by

**Our reply**

Done

**Referee comment**

12-4: *zone of aurora polaris at various altitudes*
Auroral zone? What latitudes do you mean?

**Our reply**

Here we mean latitude ~70 degree and more

**Referee comment**

12-5: *charge structures* - ?
May be replace this with "disturbances" or "irregularities"?

**Our reply**

Done

**Referee comment**

12-8: *half orbits* - Please explain where it is exactly.

**Our reply**

Corrected

**Referee comment**

12-9: *right* ->lower ?

**Our reply**

No - right . Right part of bottom figure.

**Referee comment**

12-11: The word *apparently* is repeated several times.
Eliminate *just*.

**Our reply**

Done

**Referee comment**

Figure 9: Please indicate with arrows the whistlers and the plasmospheric hisses in the figure

**Our reply**

Done

**Referee comment**
13-3: *magneto-conjugate point*
Magneto-conjugate point of what?

**Our reply**
That doesn't matter. Deleted.

**Referee comment**
13-6 and further:
It is important over which latitudes the electron precipitations of these or that energies are registered. Please be more specific about what the observed precipitations mean for the ionospheric effects (geophysically).

**Our reply**
Done

**Referee comment**
14-17: width latitude

**Our reply**
Done

We hope that after these corrections the referee will find our MS suitable for publication in *Annales Geophysicae.*

On behalf of all the authors
Sincerely Yours
Anatoliy Lobzin

---

## Referee Report (RR1)

The work describes the DIAS software for the ionosphere anomalies detection based on satellite data. As an example a complex research of the state of the medium-latitude ionosphere during a geomagnetic storm on January 20, 2010 is performed. The paper provides some indications of ionospheric anomalies coincident with the observed storm. The paper is explicit and ready for publication.
A few minor remarks.

-Abstract. Correct date. It is not January 10.
-DEMETER the satellite. The VLF band is 0.02-20 kHz.
-Figure 2. The bottom graph. How the colored bar on the right is connected with the line? The same for Figure 5.
-Figure 7. What are units of Y-axes?

---

## Author Response (AR2)

Dear Dalia Buresova,

Thank you for your comments and corrections. Please, find below the answers on your remarks:

**You say:**
The name of the SW you have designed is „Detection of Ionosphere Anomalies Software" with acronym DIAS. In your reply to the referee's comments, you stated that at the time of creating the SW, you did not know the European server with the same acronym, which could have happened. You also stated that "But, I think it is possible to remove the acronym" DIAS "from the paper", which was not done in the modified version. In the title, abstract and below in the text, you use the acronym DIAS and add "Software" (DIAS Software), which duplicates the letter "S" in the acronym. From my point of view, I would recommend using the acronym DIA and add the full word "Software" or "SW" for short, then there would be no duplication (e.g. DIA Software) and no confusion with acronym DIAS.
**Anwser:**
All acronyms "DIAS" replaced to "DIA"

**You say:**
Abstract. Correct date. It is not January 10.
**Anwser:**
Corrected

**You say:**
DEMETER the satellite. The VLF band is 0.02-20 kHz.
**Anwser:**
Corrected

**You say:**
Figure 2. The bottom graph. How the coloured bar on the right is connected with the line? The same for Figure 5.
**Anwser:**
Corrected. The bottom color bar was removed.

**You say:**
Figure 7. What are units of Y-axes?
**Anwser:**
This is a Magnetic field induction in nT. Added.

Kindest regards,
Anatoliy Lozbin